# Father Presence, Father Engagement, and Specific Health and Developmental Outcomes of Mongolian Pre-School Children

**DOI:** 10.3390/children8070584

**Published:** 2021-07-08

**Authors:** Lesley A. Pablo, Ryenchindorj Erkhembayar, Colleen M. Davison

**Affiliations:** 1Department of Public Health Sciences, Queen’s University, Kingston, ON K7L 3N6, Canada; l.pablo@queensu.ca; 2Department of International Cyber Education, Graduate School, Mongolian National University of Medical Sciences, Ulaanbaatar 14210, Mongolia; renchindorj@mnums.edu.mn; 3Department of Global Development Studies, Queen’s University, Kingston, ON K7L 3N6, Canada

**Keywords:** early childhood education, Mongolia, father involvement, health determinants, multiple indicator cluster surveys

## Abstract

This study explored father involvement as a social determinant of child health within the context of macro-environmental changes in Mongolia. Using data for children aged 3–4 from UNICEF’s Multiple Indicator Cluster Surveys, this cross-sectional analysis examined the association between father presence and engagement with child health and educational outcomes. Multivariate regression modeling was employed to identify associations between father presence, engagement, and child outcomes including fever, respiratory illness, diarrhea and preschool attendance. In unadjusted analyses, father engagement was associated with higher odds of preschool attendance (Odds Ratio (OR) = 1.12; 95% Confidence Interval (CI) 1.04–1.20) but not with child illness (OR = 1.04; 95% CI 0.95–1.14). Father engagement was no longer associated with preschool attendance after controlling for potentially confounding variables (OR_adj_ = 0.95; 95% CI 0.88–1.03). Unadjusted and adjusted analyses showed that father presence was not associated with acute illness or preschool attendance. Results also suggest that a larger proportion of children were engaged in activities by their mother compared to their father or other adults. Data indicate that father presence and engagement were not associated with child illness or preschool attendance. Factors such as maternal education, household wealth, and region of residence are stronger predictors of preschool attendance and should continue to be considered for promoting child health and development in Mongolia.

## 1. Introduction

Mongolia has experienced the effects of climate change and a dramatic socioeconomic and constitutional transition that has impacted many households, particularly those of nomadic pastoralist families [1,2,3,4]. During the 1900s, Mongolia was governed by a socialist government which provided access to important social services such as healthcare and education for all Mongolians [1,5]. Nomadic pastoralists benefited from additional services supporting their livelihoods including veterinary services and protection for livestock, water provision, and transportation for herding activities [1,2]. However, in the late 1900s, a decline in Soviet involvement as well as the political and socioeconomic transition from a communist society and centralized economy led to the dismantling of many state-provided supports [1,2,4,5,6]. This was of great concern to all Mongolians, particularly for herder families, since the change in government support coupled with increased severity of winter conditions have increasingly threatened livelihoods [1,3]. Mongolians are traditionally nomadic people where pastoralist animal herding remains an integral part of their livelihood, lifestyle, and culture. There are strong ties to Shamanism and Buddhism for most of the nomadic population, with the exception of Kazakh minorities in the western region. The male and female parent roles in households in Mongolia have historically been explicitly gendered, where men as household heads are responsible for herding and tending to animals, while women are responsible for household tasks and child-rearing [7]. Due to an increase in desertification and overgrazing in certain regions as well as significant growth in the mining industry, many herder families have been forced to live in more urban settings, thereby transforming their nomadic lifestyle [8,9]. In addition, environmental change as well as political and economic transitions brought shifts in gender roles where some households saw women become increasingly involved in the outside workforce (such as in service, finance, or trade jobs) or being involved in more animal husbandry tasks [5,10,11,12]. In other cases, men have lost their traditional livelihoods or employment, making them less of a prominent head and primary provider for the household [5,11,12]. The UN Gender Inequality Index (GII) is based on indicators of female reproductive health, empowerment, and workforce participation, and ranges from zero to one, with higher numbers denoting greater gender inequality. Mongolian GII has shown a steady decrease (improvement in gender equality) over the post transition period, with GII of 0.501 in 1995 and 0.301 in 2017 [13,14].

Currently, from a legal perspective, parenting roles in Mongolia for fathers and mothers are equivalent and for instance, parental leave is offered equally to female or male parents [15]. However, parenting still remains strongly gendered and mothers tend to take a more active day-to-day role in the lives of young people particularly for children under the age of five. Fathers are still considered to be the main financial provider and head of household due to traditional and cultural aspects of Mongolian patriarchal society (K. Sukhbaatar, Health Sociologist at Mongolian National University of Medical Sciences, personal communication, December 2020). Mongolians commonly live in multi-generational households and maintain a cultural belief that considers the family unit to be more important than any specific individuals within a family, be they child or adult [16,17]. In terms of parenting practices, Mongolian parents are motivated to build positive and necessary social attachments in their children, and also emphasize that growing autonomy is important as a way to increase agency [17,18]. Although some parents partake in corporal punishment, Mongolian parents had one of the lowest rates of corporal punishment when compared to families in other low- or middle-income countries [17,19].

Patriarchic cultural values remain ingrained in society and any shifts in gender roles due to macro-environmental changes could have effects on family dynamics and parenting practices in the household. For instance, some men or fathers who lose their identity and familial role may feel helpless or may even create an unstable family environment through alcoholism or domestic violence [16,20,21]. However, in other households, men who are unemployed may see this as an opportunity to spend more time with their children [22]. The changing roles of men and fathers have been explored in other contexts [23,24,25], however, little is known about how political and economic transition may have influenced parenting by men in Mongolia. Representatives from UNICEF Mongolia have voiced concerns regarding changing gender roles and tasks and their relation to child-related outcomes in Mongolia (U. Sereeter, ECD Specialist at UNICEF Mongolia, personal communication, February 2017).

The early years of life are a crucial time for children in terms of their physical, socio-emotional, and cognitive development [26]. These areas of development are known to be strongly influenced by the factors of social and physical environments [26,27]. Several theoretical models or frameworks have been used to understand the complex relationship between father involvement and child outcomes. One is designed around Bronfenbrenner’s Ecological Theory of Development [28,29]. This framework has often been used to depict the interactions and influences acting in nested ecological levels to explain child development and well-being [28,29]. From a child’s perspective, the model consists of the microsystem which involves direct interactions between children and parents, the mesosystem that involves interaction between parents or other family members, the exosystem which involves the parents’ relationship with other people, the *macrosystems* which are the encompassing social and environmental factors that affect the other systems (such as the characterization of gender roles or the political and economic climate), and the chronosystem which relates to influences that are historically derived or specific to a particular time period that an individual or family is living within [28,29]. Within the Mongolian context, the macro-environmental changes fall within the macro- and chronosystems and these changes can affect aspects of the mesosystem and microsystem, thereby influencing family dynamics or father-child interactions, respectively. Paternal effects on child outcomes are mainly part of the microsystem where it is argued that having several adult microsystems (i.e., Interactions with both parents as opposed to one) can benefit the child, provided that these interactions and relationships as well as interactions between parents are positive [28]. Interactions within and across systems can ultimately build social capital which is achieved when a father is warm and supportive towards their child, cooperative with the mother, and connected with others in the community such as teachers, or a child’s friends [28,30]. In addition, since mothers and fathers can exhibit different styles of interaction, children can benefit from this diversity in their microsystem interactions [28,31]. For instance, in addition to playful interactions, mothers have been shown to spend more time in caregiving or nurturing activities compared to fathers while fathers themselves spend more time in playful rather than caregiving activities [32]. While mothers and fathers can both engage in playful activities, scholars have suggested that fathers have unique types of playful interactions such as ‘rough-and tumble’ play and teasing interactions that can be beneficial for child development [28,33]. This idea is further supported by a recent meta-analysis by Jeynes [34] which showed that fathers’ contributions towards child well-being is distinct from that of the mother. Here, father involvement showed a distinctly positive association with measures such as child psychological, social, and academic outcomes [34]. The underlying mechanisms at work can be described using the ‘positive father involvement’ construct proposed by Lamb, Pleck, and colleagues [28,35]. They postulate that ‘involvement’ is an umbrella term which describes whether the father is (1) accessible to their child and is therefore present to address the child’s needs, (2) engaging with their child, allowing them to identify their child’s needs and engage them in different activities, and (3) responsible for their child by planning and managing the needs of their child. This suggests that paternal supervision and indirect involvement such as financial contributions can be beneficial to the child [29,30].

Understanding the role of the Mongolian father will help contribute towards knowledge regarding factors associated with child health and well-being in Mongolia. It may also be beneficial to understand how fathers play a role in issues highlighted in the United Nation’s Sustainable Development Goals (SDGs) such as those that highlight the need to focus on communicable diseases and access to education by 2030 [36]. Key targets include 3.3 “end[ing] the epidemics of […tuberculosis, …hepatitis], and water-borne diseases…” [36] and target 4.2 “ensur[ing] that all girls and boys have access to quality early childhood development, care and pre-primary education so that they are ready for primary education” [36]. An area of particular interest is early childhood education. ‘School readiness’ is an important aspect of child development since attending an organized institution such as preschool helps equip children with the cognitive and socio-emotional skills that are necessary to be successful in school [37]. Moreover, preschool can help facilitate the transition from a home-environment to a school environment for both the children and their parents or caregivers [37]. Benefits have been demonstrated in studies conducted globally where children who attend a school readiness program show improved cognitive outcomes such as language (e.g., speaking or writing), problem-solving, and learning skills, are more prepared for primary school, and are more likely to complete high school [38,39,40,41,42]. In Mongolia, data suggests that preschool attendance was approximately 68% in 2013 [43].

Studies from low-, middle- and high-income countries that involve the father have examined the father’s role and its effect on cognitive, academic, and socio-development outcomes in children [29,44,45,46,47,48,49]. For instance, in low- and middle-income countries, high paternal stimulation and involvement is associated with improved language and literacy skills in children [45,48]. Supportive behaviours from the father has also been shown to be positively associated with their child’s emotional development as shown in a high-income country context [46]. In contrast, low levels of father engagement has been shown to be associated with higher aggressive and externalizing behaviour problems in children living in the United Kingdom [50]. Many other studies from developing and post-transition countries revolve around parental caregiving or engagement as well as its association with cognitive and socio-emotional outcomes in children [27,45,48,51,52,53] while others generally focus on early childhood education attendance without considering parental engagement as a determining factor [54,55]. In terms of the association between father involvement and preschool attendance, a study involving Caribbean countries found that preschool attendance was positively associated with father’s social engagement in the Dominican Republic and with father’s cognitive and social engagement in Suriname [56]. In addition, the study suggests some associations between preschool enrollment and paternal and maternal engagement in social or cognitive activities which seem to vary between parents and across countries. In post-transition settings, preschool attendance as well as the hourly length in which children attended preschool was positively associated with higher parenting quality index (i.e., the number of activities that the parents engaged in with the child) in countries such as Kyrgyzstan, Tajikistan, and Uzbekistan [57,58].

As for acute illness, research on the relationship between father involvement and risk of illness would also be beneficial since the leading causes of morbidity in Mongolian children aged 1–4 years in 2014 were diseases affecting the respiratory and digestive systems where prevalence rates were approximately 57% and 9%, respectively [59]. Many studies focus on father presence as a determinant of acute illness and suggest that children who do not live with both their biological parents have greater odds of poor health, diarrhea, and asthma [44,60,61,62]. However, studies on acute illness and physical health involve children from Western populations and mainly focus on father presence in the household [60,61,63,64,65,66]. 

There are no studies to date that explore the direct relationship between father involvement and outcomes such as school readiness and acute illness in children within the Mongolian context. The main objective of this study is to explore father presence, father engagement, and the association with preschool attendance and acute illness in Mongolian children aged 3–4 years.

## 2. Materials and Methods

### 2.1. Data Source

This cross-sectional study uses health indicator data collected through the 2013 Mongolian Multiple Indicator Cluster Surveys (MICS) designed by UNICEF and conducted in collaboration with the National Statistical Office of Mongolia (NSO) [43,67]. Mongolia experienced rapid mining related economic growth between 2008–2014, and the employment rate peaked in 2013. Due to rapid development, there were few family-oriented housing options near the mine sites. This led to temporary migration of mostly men for mining work in ger camps. Although older, the 2013 data were collected at a critical period and remain crucial to our understanding of the influence of these kinds of societal and economic changes on father’s presence and child health outcomes in this time period.

MICS surveys are used to collect health indicator data at semi-regular intervals for women and children in developing nations to support efforts to monitor country- and region-specific progress towards goals in maternal and child health, including those outlined in the Millennium Development Goals and SDGs [36,43]. Self-reported data on various health indicators were collected through structured interviews at the household and individual level and are publicly available upon request. Data were collected through the household, women’s, and child’s questionnaires. All questionnaires in each household are linked through cluster and household identification codes. 

### 2.2. Study Sample

A multi-stage, stratified cluster sampling method was used to ensure that data were collected from a nationally representative sample of the Mongolian population [43]. Sampling was primarily based on the geographic location of households (i.e., region of residence and urbanicity). The mothers or caregivers of 2374 male and female children aged 3–4 completed the ‘Children Under Five Questionnaire’ and 95.4% of respondents to the questionnaire were the child’s mother. Children were excluded from the study if their fathers were deceased or their status was listed as ‘don’t know’. Children were also excluded if their fathers were their primary caretakers; these cases were excluded since the father’s education may be reported as ‘Father’s Education’ as well as ‘Mother’s or Caretaker’s Education’. In addition, in households with more than one child aged 3 or 4, the oldest child was selected while the remaining children were excluded from analysis. For analyses on father presence, the resulting unweighted sample size for children aged 3–4 was 2220. Analyses on father engagement only included children whose fathers lived in the household and excluded those with missing data on father’s education, resulting in an unweighted sample size of 1896. Response rates for the ‘Household Questionnaire’ and the ‘Children Under Five Questionnaire’ in 2013 were greater than 90% [43]. It is important to note that “maternal” factors refer to the child’s mother or primary “maternal-like” caretaker (i.e., grandmother, aunt, etc.).

### 2.3. Measures

The father’s role was the primary exposure and was measured using two indicators: father presence in the home and level of father engagement. Information on father presence was available for all children under age 5 and was obtained through the ‘Household Listing Questionnaire’. Heads of the household or their designate were asked whether the child’s biological father was alive, and if so, whether he lived in the household at the time of the survey. Information on level of father engagement was only available for children aged 3–4 years and was obtained through the ‘Children Under Five Questionnaire’. Mothers or caretakers of the child were asked whether any adult(s) aged 15 and over engaged in six listed activities in the three days prior to the survey and to indicate whether the adult was the child’s father. Activities include whether the father ‘read books or looked at picture books’, ‘told stories’, ‘sang songs’, ‘took the child outside the home’, ‘played with’, or ‘named, counted, or drew things’ to or with their child. These activities are indicators of cognitive and socio-emotional caregiving provided to the child by their parents [51]. Similar to previous studies [45,53], the level of father engagement was determined by summing the number of activities through which the father engaged with their child, ranging from 0–6. Indicators of father engagement were used under the assumptions that each item was equally weighted with respect to their contribution to a child’s cognitive and socio-emotional well-being and that father-child interactions were generally positive. Indicators of cognitive and socio-emotional engagement have been tested in a previous study by Bornstein and Putnick [51] where the cognitive measures had a Kuder-Richardson 20 reliability score of 0.68 while the socio-emotional measures had a score of 0.64. Similarly, Jeong and colleagues [45] showed that the same measures of father engagement had good internal consistency (α = 0.77) in a study involving several LMICs.

Child outcomes were measured using two indicators from the Children Under Five Questionnaire: acute illness and preschool attendance. Acute illness was dichotomized into the child being ill or not ill in the two weeks prior to the survey. Mothers or caretakers were asked whether the child had diarrhea, was ill with a fever, or showed symptoms of an acute respiratory infection in the two weeks prior to the survey. A child was classified as being ill if they were reported to show any symptoms of acute illness. As for school readiness, preschool attendance was used as an indicator where respondents were asked whether the child ‘attends any organized learning or early childhood education programme’. In order to retain as many cases as possible, variables with “DK” (Don’t know) responses were recoded into “No” responses (*n* = 3).

Other potentially confounding variables and key covariates were identified and adjusted for in the analyses based on previous studies. These include child’s age, maternal education level, paternal education level, household wealth quintile, region (Figure 1) and type (urban or rural) of residence, number of children’s books in the household, and number of adults living in the household [45,51,53]. Household wealth quintiles were provided in the dataset and were based on scores calculated through principal components analysis using information on household assets, dwelling characteristics, water facilities, and urban/rural residence [43]. Analyses also adjusted for the early childhood development index (ECDI) score (from 0–11) and is based on the number of positive responses by the mother or caretaker to questions regarding literacy-numeracy, socio-emotional, and learning skills that have been previously validated and used in other studies [45,48,68]. Questions include whether the child can recognize simple shapes, count, or get along well with others [43,48]. 

### 2.4. Data Analysis

All analyses were performed using SAS Studio University Edition v.9.4 (SAS Institute Inc., Cary, NC, USA). Unadjusted and adjusted multivariate logistic regression modelling was used to identify associations between father presence, level of father engagement, and child outcomes while accounting for potential confounding and the effects of other covariates. The PROC LOGISTIC procedure from SAS v.9.4 was used to obtain final models through the backwards selection procedure. Variables were chosen according to a liberal cut-off value of *p* < 0.15. In order to account for the multilevel clustered nature of the data, error estimates were adjusted using the PROC SURVEYLOGISTIC procedure in SAS v.9.4. Cochran’s Q tests were also performed to compare the involvement of any household adults in each type of activity with the child. Cases with missing values in any of the exposure, outcome, or covariate variables were excluded from analyses and accounted for 0.16% of all children aged 3 or 4 with complete questionnaires. 

All analyses were performed using individual sample weights. The normalized weights were provided for all children under 5 to adjust for regional differences in sampling probabilities as well as for questionnaire non-response [43].

## 3. Results

Table 1 details the characteristics of the study population.

### 3.1. Acute Illness

Table 2 shows results from unadjusted logistic regression models for father presence and acute illness as well as for level of father engagement and acute illness. Compared to children whose fathers did not live in the household at the time of the survey, children who lived with their fathers had a slightly higher odds of having an acute illness (Odds Ratio (OR)_unadj_ 1.06; 95% Confidence Interval (CI) 0.72, 1.56), however, this was not statistically significant. Similarly for father engagement, an increase in the level of engagement indicated slightly higher odds of acute illness in children (OR_unadj_ 1.04; 95% CI 0.95, 1.14) compared to children without engaged fathers, however, this too was not statistically significant. This suggests no statistically significant association between father presence or level of father engagement and acute illness in children in 2013.

### 3.2. Preschool Attendance

Table 2 shows results from unadjusted and adjusted logistic regression models involving father presence, level of father engagement, and preschool attendance in children aged 3–4. For father presence, bivariate analysis suggests no association between fathers living in the household and the odds of preschool attendance in children (OR_unadj_ 0.79; 95% CI 0.60, 1.04). This was similarly shown in analyses that adjusted for potentially confounding variables and key covariates (OR_adj_ 0.91; 95% CI 0.66, 1.26). As for level of father engagement, bivariate analyses suggests that higher levels of father engagement are associated with higher odds of preschool attendance (OR_unadj_ 1.12; 95% CI 1.04, 1.20), however, this association was no longer significant after adjusting for potentially confounding variables and covariates (OR_adj_ 0.95; 95% CI 0.88, 1.03).

### 3.3. Factors Associated with Preschool Attendance

Table 3 shows the variables included in the final regression models for measuring the association between father presence (M1) and engagement (M2) with preschool attendance. Notable variables from both models that are associated with preschool attendance include child’s age, number of adults living in the household, mother’s education level, household wealth quintile, and region of residence. Children aged 4 have higher odds of attending preschool compared to children aged 3 (OR_M1_ 1.93, 95% CI 1.55, 2.40; OR_M2_ 1.54, 95% CI 1.21, 1.97). The number of adults in the household also had an effect where a higher number of adults meant lower odds of preschool attendance (OR_M1_ 0.83, 95% CI 0.72, 0.94; OR_M2_ 0.78, 95% CI 0.67, 0.92). As for mother’s education level, lower educational attainment was associated with lower odds of preschool attendance in their children. For instance, children whose mothers had a primary education level had lower odds of attending preschool compared to children whose mothers had a college or university education (OR_M1_ 0.36, 95% CI 0.23, 0.56; OR_M2_ 0.44, 95% CI 0.26, 0.74). Similarly, household wealth quintile was also associated with preschool attendance. Children living in the richest quintile had 13.5 times the odds of preschool attendance in Model 1 and 11.4 times the odds of preschool attendance in Model 2 when compared to the odds of children from the poorest quintile (M1 95% CI 7.94, 22.82; M2 95% CI 6.24, 20.97). Preschool attendance also varied by region of residence. For example, children living in the Eastern region had 3.7 times the odds of preschool attendance in Model 1 and 5.8 times the odds of preschool attendance in Model 2 compared to the odds of children living in Ulaanbaatar (M1 95% CI 2.24, 6.22; M2 95% CI 3.28, 10.30).

In addition to these factors, Model 2 also adjusts for early childhood development index (ECDI) score and paternal education attainment. A higher development score was associated with higher odds of preschool attendance (OR 1.32, 95% CI 1.22, 1.42). Similar to maternal education attainment, children whose fathers had lower paternal education level, such as primary education, had lower odds of preschool attendance compared to children whose fathers had a college or university education (OR 0.53, 95% CI 0.31, 0.89).

Tests for possible multi-collinearity were also performed and none of the variables were highly correlated. As expected, there were moderate correlations amongst the wealth index and parental education variables, the maternal and paternal education variables, as well as between the region of residence and urbanicity variables. However, based on previous literature, these were still included in the model due to their unique possible confounding effects in the relationship of interest.

### 3.4. Adult Engagement by Type of Activity

Figure 2 displays the types of activities that the mother, father, and other adult over age 15 in the household engaged in with children aged 3–4 years. It can be seen that a higher proportion of children were reported to be engaged in activities by their mothers compared to fathers and other adults in the household. For each activity, results from Cochran’s Q tests suggest that proportions of children aged 3–4 who were engaged in activities with adults in the household differed by the adult who engaged with them (*p* < 0.0001). A higher proportion of children were engaged in activities with their mothers compared to other household adults with values ranging from 29–48% for mothers, 14–35% for fathers, and from 12–25% for other household adults.

## 4. Discussion

Using data from 2013, this study shows that father presence and father engagement with their children are not directly associated with acute illness and preschool attendance. Unadjusted results suggest an association between father engagement and preschool attendance, however, adjusted results indicate that other demographic factors are more associated with preschool attendance. Parents and adults in the household vary in terms of the amount of engagement with children. Mothers also had higher reports of engagement in activities with children compared to fathers and other adults in the household.

### 4.1. Acute Illness

From this data, there does not seem to be an association between father presence, engagement, and acute illness in children. There are several complex factors that are known to contribute to the risk of diarrhea and respiratory illness in children. For instance, known risk factors for diarrhea include having an unprotected water source, regular contact with contaminated feces or domesticated animals, exposure to butchering of animals, unsanitary food handling, and poor handwashing practices, and is a concern particularly for those living in nomadic pastoralist households [69,70]. While these may increase the risk of diarrhea in children, this may be attenuated by improved adult supervision. Increased supervision of the child through close proximity and keeping them in clear sight has been shown to reduce the risk of unintended injury in children [71]. A similar mechanism can be applied for protecting against acute illness where engaging and interacting with children allows adults like the father to monitor what the child is touching or eating. Having an extra parent or adult in the household to supervise children can be very beneficial [72]. Parents who are aware that children came into contact with animals or dirt while engaged in activities such as playing or going outside the home may remind their child to wash their hands before eating or after defecation or they may monitor their contact with animals (such as dogs) or feces. However, this may not be completely effective if their primary water source is contaminated or if the children are fed uncooked or contaminated animal products [70]. Parental supervision can similarly be applied for protecting against acute respiratory illnesses where improved supervision can help reduce the child’s risk of illness. However, Ulaanbaatar is currently facing a major outdoor air pollution problem and is home to most cases of acute respiratory illness in Mongolian children [43,73]. While parents may help reduce their child’s risk of illness by ensuring they wear good quality face masks or wash their hands, children will continue to be at risk for diseases as long as the air pollution issue persists in Mongolia [73,74,75,76].

### 4.2. Preschool Attendance

Other factors seem to be more strongly associated with preschool attendance in children than father presence and engagement. For instance, similar to the results found in other studies conducted in both western and post-transition settings, children who are older are more likely to attend preschool compared to younger children [54,77,78]. Living in households with several adults is associated with lower odds of preschool attendance and may be due to the availability of adult caregivers, such as grandparents, to care for the children [54,78]. As expected, additional factors such as parent’s education level, household wealth quintile, region, and type of residence were found to be determinants of preschool attendance in Mongolia.

A high proportion of well-educated respondents in the study is characteristic of the Mongolian population. Parental education level is known to be an important contributor to child development where parents with a high level of education may be more likely to see the benefits of preschool for their children as well as provide them with positive and stimulating environments [45,48,78,79]. Therefore, results showing that higher levels of maternal education are associated with higher odds of preschool attendance were expected and consistent with other studies, including one performed in the post-transition countries of Georgia and Kazakhstan [54,78]. In addition, while parental education in general was shown to be positively associated with preschool attendance, this study showed that maternal education had a stronger relation to preschool attendance compared to paternal education. This was expected since maternal education is generally more strongly associated with level of early child development and parental support for learning compared to paternal education in low- and middle-income countries [48].

Household wealth quintile is also seen to play a role in preschool attendance. Preschools in Mongolia are provided without cost to families, however, parents are expected to pay out of pocket for higher quality daily meals as well as school supplies [80]. These charges were found to be inequitable where low-income households were expected to pay the same fees as high-income households [80]. With this current system in place, it may explain why children living in wealthier households have greater odds of attending preschool compared to those in more disadvantaged households.

Important regional differences were also found in this study, particularly when comparing attendance in Ulaanbaatar, the nation’s capital city, with that in other regions. The finding that children living outside of Ulaanbaatar have higher odds of attendance compared to those living in the city was unexpected since the capital city is presumed to have improved access to education, and therefore, should translate into a higher attendance rate compared to other regions. One likely explanation for this is the fact that during the transition period, many Mongolians, including former herders, internally migrated from rural to urban areas such as Ulaanbaatar in search of employment as well as better access to health and education services [4,81,82]. Today, close to half of the total Mongolian population lives in Ulaanbaatar and are thereby putting a strain on the education system in the city [80,82]. A report by the World Bank explains that kindergartens in Ulaanbaatar are filled to capacity with many preschool-aged children finding themselves on long waitlists [80]. This lack of access to preschool may explain why living outside of Ulaanbaatar is associated with increased enrollment in preschool (Dr. E. Tsogzolbaatar, Department of Epidemiology and Biostatistics, Mongolian National University of Medical Sciences, personal communication, June 2019).

### 4.3. Adult Engagement by Type of Activities

Results show that a larger proportion of children were engaged in the six activities with their mother compared to their father or other adults living in the household. This was expected since mothers are usually the primary caregivers of children, and in Mongolia, households typically have gendered roles where mothers are responsible for child care [11]. These results are consistent with another study where mothers from LMICs in Central and Eastern Europe, South Asia, East-Asia and the Pacific, sub-Saharan Africa, the Middle East, and the Caribbean were generally reported to engage with children in activities more than the fathers [53]. This may be due to fathers being less likely to engage in activities such as reading if the mother is already interacting with the child. In addition, the authors found that in households where mothers completed a high level of education, fathers were less likely to engage with children [53]. One possible explanation is that more educated mothers have greater skills and awareness regarding childhood development type activities [79]. The mothers may also have high parenting expectations and could subsequently moderate the activities between children and their fathers through a process known as ‘maternal gatekeeping’, especially when mothers deem fathers incapable of caring for children [83,84].

### 4.4. The Father’s Role in Child Well-Being

While this study showed no association between father presence and engagement with acute illness and preschool attendance in Mongolian children after adjusting for demographic factors, previous studies have shown that positive involvement from fathers is an important determinant in other child well-being and development outcomes. Positive father involvement has been shown to be positively associated with outcomes such as language skills, literacy skills, and emotional development in children [45,46,48]. Therefore, it is important to note that the lack of a statistical association between father involvement and child outcomes (specifically acute illness and preschool attendance) should not be interpreted as evidence that fathers are not an important determinant of child health and well-being. In addition, based on Bronfenbrenner’s Model, it is worth noting that the father’s influence on a child’s well-being is not an independent entity and is rather influenced by the family system and the macro-environmental conditions in which the father and child are situated. Parenting approaches and motivations are socially and culturally bound [85,86] and the influence of Mongolian fathers on the outcomes of their preschool children cannot be understood separately from an understanding of the broader context and the role of fathers in that context. Mongolians are traditionally nomadic herding peoples however, desertification, overgrazing and a growing mining industry in Mongolia has ultimately led to an increase in urbanization of families in some regions of the country and a general transformation of the nomadic lifestyle [8,9]. Historically, responsibility for herding animals and relevant hard-labor duties have been largely born by men as the household heads, whereas women are gendered for household duties and parenting, including in more recent times supporting the formal education of children [7]. Although Mongolia is considered to have a relatively high level of gender equality as compared to other countries at a similar human development index position, patriarchist practices remain predominant, including more managerial positions and political power assumed by men whilst women have caregiver roles and often a higher level of formal education [87]. In this context, it might not be expected that fathers would have an influence on acute illness or preschool readiness, and particularly not equally across boys and girls.

Fathers are known to have unique types of playful interactions such as ‘rough-and tumble’ play and teasing interactions with children [28,33]. These kinds of father interactions are prevalent in traditional and contemporary Mongolian culture. In the past, and currently in many families, fathers teach young children to ride and care for horses, a cultural lesson as well as a life skill. Mongolian fathers also often accompany their children to summer festivals where traditional cultural activities are highlighted or they visit neighboring families with children to build familial social networks. These kinds of parental activities demonstrate the importance of fathers for gender role-modelling (particularly for boys) as well as cultural teaching and relational development for both boy and girl children. Again, since the emphasis in these types of father interactions in Mongolia are relationship development and the establishment of gender roles, father interaction might not be expected to affect child health or preschool attendance. While this study tried to utilize the comprehensive MICS dataset and the engagement and outcome measures included in it, further study could be undertaken to fully understand the diverse influence of fathers on preschool children in Mongolia. It is important to consider the impacts of a transitional environment on father-child interactions more broadly as well as impacts on cultural teaching, relationship development, familial networks and the subsequent effects on other child outcomes.

### 4.5. Limitations

This is an exploratory study that provides a snapshot of the relationships between father presence and engagement and child-outcomes in a rarely studied population and helps to highlight directions for further scholarship. The cross-sectional nature of this study means that only associations rather than causal inferences can be drawn. Since surveys provide a snapshot of certain indicators around the time of the survey, it may not necessarily reflect what is true for a child or household throughout all seasons of the year. The use of caregiver-reported data as well as questionnaire design may introduce information bias. Caregivers were assumed to be able to accurately report the data on all variables including presence of symptoms of acute illness in their child in the last two weeks. Respondents for the children under five questionnaire were in most cases the mother of the child. Response bias could have arisen if the mother underestimated or provided inaccurate accounts of whether the father engaged with the child, particularly if there is parental conflict [88]. An additional source of misclassification is the use of six activities as a measure of father engagement since they did not capture other forms of engagement such as feeding or bathing the child or the length or quality of these engagements. Moreover, the questions might not completely capture the specific details of Mongolian parenting practices. This was mainly due to the survey design where questions on engagement were intended to specifically measure parental support for learning rather than overall parental involvement. Father presence and engagement are obviously very important for child development. We were not able to rule out a potential lack of association that may be due to the scale attenuation (ceiling) effect. Overall, 54.5% of children engaged with fathers in at least one activity. We were not able to determine a mechanism or threshold of any effect with the measure we had available to us. Further study with different measures would potentially contribute additional understanding. Child interaction with “father-figures” (such as step-fathers or grandfathers) was also not measured. This is an area that could also have further investigation since previous studies have shown that different family structures influence child outcomes in diverse ways [60,89]. The region of residence and urbanicity were measured only at the time of the survey but do not capture whether the family recently moved. In terms of father presence, details on periodic absences or length of absence, if any, were also not captured. This study used data from the MICS 2013 survey period which was the most recent dataset that contained some measure of father involvement as well as key demographic variables required for this analysis.

## 5. Conclusions

Father presence and father engagement were not found to be associated with preschool attendance or acute illness in preschool aged children in this study using MICS 2013 data. Instead, other factors such as maternal education, household wealth quintile, and region of residence are stronger predictors of school readiness and should continue to be considered when looking to improve the well-being of Mongolian children. However, the effects of the father on other outcomes such as cognitive and socio-emotional development still remain to be evaluated. Further studies looking at the quality of father–child relationships and the relationship between children and other adults in the household in Mongolia are still warranted. Because future developments in Mongolia will also likely focus on the mining sector, particularly in the South Gobi, we believe there is value in exploring the influence of these developments, and father’s potential mining employment, on child and family outcomes. This was a valuable exploratory study of the role of fathers in determining aspects of child health and preschool attendance in post-transition era Mongolia.

## Figures and Tables

**Figure 1 children-08-00584-f001:**
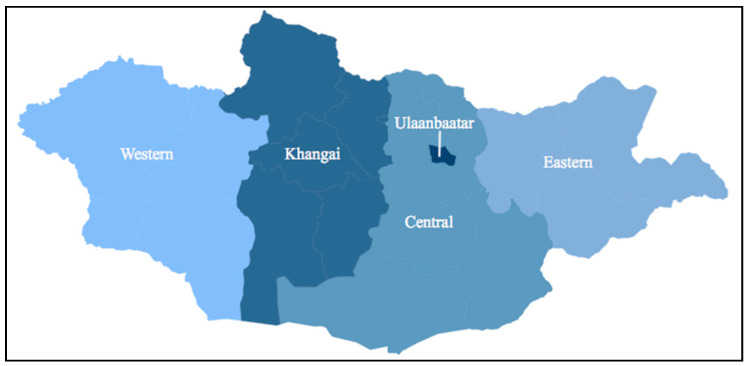
Regional map of Mongolia.

**Figure 2 children-08-00584-f002:**
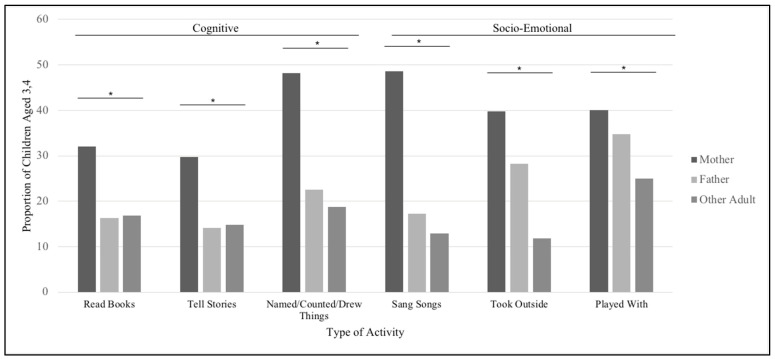
Proportion of children aged 3–4 who were engaged by a parent/adult in the three days prior to the survey by type of cognitive and socio-emotional activity. ‘Other adult’ refers to any adult aged 15 and over who is not the child’s father or mother and is a member of the household. * *p* < *0*.0001; obtained from Cochran’s Q tests comparing proportions between parents/adults for each type of activity.

**Table 1 children-08-00584-t001:** Characteristics of the study population by primary exposure and outcome (unweighted).

		Father Presence		Father Engagement ^a^		Acute Illness ^c^		Preschool Attendance ^c^	
Sample Characteristics		*n*	Mean (SD)/%	*n*	Mean (SD)/%	Yes (%)	No (%)	Yes (%)	No (%)
Total Sample		2220		1896		12.2	87.8	68.6	31.4
Child’s Age (years)									
	3	1090	49.1	937	49.4	58.4	48.2	45.8	57.4
	4	1130	50.9	959	50.6	41.6	51.8	54.2	42.6
Child’s Sex									
	Male	1108	49.9	936	49.4	54.6	48.6	49.2	49.7
	Female	1112	50.1	960	50.6	45.4	51.4	50.8	50.3
Early Childhood Development Score		-	-		6.7 (1.7)	-	-	-	-

Number of Children’s Books in Household			3.8 (3.7)		3.8 (3.7)	-	-	-	-

Number of Adults aged 18+ in Household			2.3 (0.8)		2.3 (0.7)	-	-	-	-

Number of Children aged 0–4 in Household			1.4 (0.5)		1.4 (0.6)	-	-	-	-

Maternal/Caretaker Education									
	None	136	6.1	116	6.1	6.5	6.1	3.8	11.1
	Primary	167	7.5	151	8.0	6.9	8.1	4.4	15.8
	Lower Secondary	376	16.9	327	17.2	13.4	17.8	13.6	25.2
	Upper Secondary	533	24.0	466	24.6	26.4	24.3	23.8	26.2
	Vocational	183	8.2	141	7.4	9.1	7.2	7.7	6.9
	College/University	825	37.2	695	36.7	37.7	36.5	46.6	14.9
Paternal Education ^b^									
	None	-	-	231	12.2	12.6	12.1	7.8	21.6
	Primary	-	-	243	12.8	13.8	12.7	8.9	21.3
	Lower Secondary	-	-	410	21.6	16.9	22.3	20.4	24.3
	Upper Secondary	-	-	379	20.0	21.2	19.8	21.6	16.4
	Vocational	-	-	186	9.8	12.1	9.5	10.8	7.7
	College/University	-	-	447	23.6	23.4	23.6	30.5	8.6
Wealth Index									
	Poorest	538	24.2	493	26.0	23.4	26.4	14.2	51.8
	Second	471	21.2	380	20.0	23.4	19.6	19.8	20.5
	Middle	434	19.6	363	19.2	17.8	19.3	22.6	11.6
	Fourth	376	17.0	321	16.9	16.0	17.1	19.8	10.7
	Richest	401	18.1	339	17.9	19.5	17.7	23.6	5.4
Region									
	Western	366	16.5	339	17.9	16.9	18.0	15.0	24.2
	Khangai	483	21.8	421	22.2	27.3	21.5	20.8	25.3
	Central	413	18.6	351	18.5	16.9	18.7	19.1	17.3
	Eastern	289	13.0	238	12.6	10.0	12.9	14.5	8.2
	Ulaanbaatar	669	30.1	547	28.8	29.0	28.8	30.6	25.0
Urbanicity									
	Urban	1249	56.3	1017	53.6	54.6	53.5	61.2	37.2
	Rural	971	43.7	879	46.4	45.4	46.5	38.8	62.8

^a^ Only includes children whose fathers were living in the household at the time of the survey. ^b^ Only available for children whose fathers were living in the household at the time of the survey. ^c^ Total sample size is 1896. Percentage sums that do not add up to 100 are due to rounding.

**Table 2 children-08-00584-t002:** Odds ratio estimates from the unadjusted and adjusted analyses of the association between the father’s role and child outcomes.

			Acute Illness			Preschool Attendance					
			Unadjusted			Unadjusted			Adjusted		
Exposure		*n*	OR	(95% CI)	*p*-Value	OR	(95% CI)	*p*-Value	OR	(95% CI)	*p*-Value
Father Presence											
	Yes	1880	1.06	(0.72, 1.56)	0.763	0.79	(0.60, 1.04)	0.095	0.91 ^a^	(0.66, 1.26)	0.560
	No	310	1.00	-		1.00	-		1.00	-	
Level of Father		1879	1.04	(0.95, 1.14)	0.392	1.12	(1.04, 1.20)	0.002	0.95 ^b^	(0.88, 1.03)	0.242
Engagement											

Note: Results from multivariate logistic regression models using survey sample weights and robust error estimates. ^a^ Adjusts for level of father engagement, child’s age, number of children’s books, number of adults aged 18 and over living in household, maternal education attainment, household wealth quintile, region of residence, and urban/rural residence. ^b^ Additionally adjusts for early childhood development score and paternal education attainment.

**Table 3 children-08-00584-t003:** Odds ratio estimates from multivariate regression models to measure the association between father presence and preschool attendance (Model 1) as well as level of father engagement and preschool attendance (Model 2).

			Preschool Attendance		
Exposure Variables			OR	(95% CI)	*p*-Value
**Model 1**					
	Father Presence				
		Yes	0.91	(0.66, 1.26)	
		No	1.00		
	Other Variables				
	Father Engagement		0.98	(0.91, 1.06)	
	Child’s Age (Years)		1.93	(1.55, 2.40)	***
	Number of Children’s Books		1.09	(1.05, 1.13)	***
		in Household			
	Number of Adults in		0.83	(0.72, 0.94)	**
		Household			
	Maternal/Caretaker Education Level				
		None	0.53	(0.31, 0.91)	*
		Primary	0.36	(0.23, 0.56)	***
		Lower Secondary	0.51	(0.34, 0.75)	***
		Upper Secondary	0.52	(0.38, 0.72)	***
		Vocational	0.61	(0.40, 0.93)	*
		College/University	1.00		
	Wealth Quintile				
		Richest	13.46	(7.94, 22.82)	***
		Fourth	6.77	(4.24, 10.80)	***
		Middle	6.45	(4.24, 9.81)	***
		Second	4.14	(2.85, 6.02)	***
		Poorest	1.00		
	Region				
		Western	2.57	(1.62, 4.05)	***
		Khangai	2.60	(1.75, 3.85)	***
		Central	2.74	(1.77, 4.22)	***
		Eastern	3.73	(2.24, 6.22)	***
		Ulaanbaatar	1.00		
	Urban/Rural Residence				
		Rural	0.96	(0.65, 1.40)	
		Urban	1.00		
**Model 2**					
	Level of Father Engagement		0.95	(0.88, 1.03)	
	Other Variables				
	Child’s Age (Years)		1.54	(1.21, 1.97)	***
	Number of Children’s Books in		1.04	(0.99, 1.08)	
		Household			
	Number of Adults in Household		0.78	(0.67, 0.92)	**
	Early Childhood Development Score ^a^		1.32	(1.22, 1.42)	***
	Maternal/Caretaker Education Level				
		None	0.65	(0.35, 1.22)	
		Primary	0.44	(0.26, 0.74)	**
		Lower Secondary	0.53	(0.34, 0.83)	**
		Upper Secondary	0.51	(0.36, 0.74)	***
		Vocational	0.74	(0.44, 1.23)	
		College/University	1.00		
	Paternal Education Level				
		None	0.48	(0.28, 0.84)	**
		Primary	0.53	(0.31, 0.89)	*
		Lower Secondary	0.73	(0.45, 1.18)	
		Upper Secondary	0.59	(0.38, 0.92)	*
		Vocational	0.58	(0.34, 0.97)	*
		College/University	1.00		
	Wealth Quintile				
		Richest	11.44	(6.24, 20.97)	***
		Fourth	6.18	(3.65, 10.45)	***
		Middle	6.96	(4.38, 11.07)	***
		Second	4.09	(2.70, 6.18)	***
		Poorest	1.00		
	Region				
		Western	3.05	(1.81, 5.16)	***
		Khangai	3.15	(1.99, 4.98)	***
		Central	3.00	(1.81, 4.98)	***
		Eastern	5.81	(3.28, 10.30)	***
		Ulaanbaatar	1.00		
	Urban/Rural Residence				
		Rural	0.81	(0.52, 1.25)	
		Urban	1.00		

Note: * *p* < *0*.05, ** *p* < *0*.01, *** *p* < *0*.001. Results obtained from multivariate logistic regression models using sample survey weights and robust error estimates. ^a^ Early childhood development index was measured as sum of positive responses/outcomes to questions regarding the child’s literacy-numeracy, socio-emotional, and learning development. Scores ranged from 0–11. This was similarly performed by [48].

## Data Availability

Data are available for research and teaching purposes upon request. Information can be found and access requests can be made at: https://mics.unicef.org/surveys (accessed on 12 September 2017).

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
