# Peer review of "Father Presence, Father Engagement, and Specific Health and Developmental Outcomes of Mongolian Pre-School Children"

_children, 2021, doi:10.3390/children8070584_

Round 1

Reviewer 1 Report

The problem discussed in the article is at the meeting point of several sciences: demography, social policy, medicine - healthcare, psychology. The idea stems from research of world organizations and institutions (the United Nations, The World Bank, National Statistical Office of Mongolia [NSO], United Nations Children's Fund [UNICEF]), analyzes of regulations, e.g. the Mongolian Law of Family, (1996), demographic analyzes, and others.

The title of the article indicates that the main focus of attention is on the changing roles of fathers in Mongolia after the systemic transformation. The literature review presented in the article indicates that although changes in male roles, including the role of the father, were studied in various socio-cultural contexts, in various economic systems (also in the context of the development of information technology, contributing to the weakening / disappearance of "boundaries" geographical and civilization in the world) - there are no such in the world literature that would include modern Mongolia. Although little is known from the scientific literature about how the political and economic transformations in Mongolia - at the turn of the 20th and 21st centuries - affected paternal roles in Mongolia, the review of other studies presented in the article provided the authors of the article with the basis for formulating specific expectations for this topic. These expectations take into account the economic situation of different societies classified on the basis of income. It is worth paying attention to the cultural tradition of the Mongols: for example, the dominant religious systems, which may be important for gender roles, including parental ones, as well as for the expectations of social egalitarianism. (There is a very interesting thread for a researcher of psychosocial phenomena. Many questions arise, for example the following question: can the identified differences between regions be related to a different worldview of the inhabitants of these regions? Did the migrations in the 21st century limited these differences? or influenced on  larger differences within regions?)

The authors drew attention to two issues: the health condition of children aged 3-4 and their emotional and cognitive development. They were treated as predictors of the child's preparation for school - his success in further education. This is important due to the development prospects of the Mongolian society. However, as the authors emphasize, their main goal was to examine the father's involvement in the relationship with the child and the relationship of this involvement with attending kindergarten and acute diseases of Mongolian children aged 3-4. The goal formulated in this way can be considered achieved. Authors' considerations on social changes for the functioning of families and the threads related to the psychological development of the child allows the authors to conclude about the impact of the activity of a Mongolian man in contact with a child on the child's health condition and its psychological development - only indirectly.

In the analysis of somatic health of young children, the indicators obtained mainly from mothers / available caregivers / children were taken into account. Data on the father's involvement in contacts with the child also came from the same sources. Similarly, data on the presence of children in kindergarten. Two questionnaires were used: 'Household Listing Questionnaire' and 'Children Under Five Questionnaire'. The data collected primarily and in this way were compared with the analyzes of other authors from research in other societies - which results from the lack of research in the Mongolian population. Expectations regarding the importance of the father and the role of preschool education in cognitive, emotional and social development were rooted in psychological studies by other researchers (Bornstein and Putnick, 2012) and documents of international organizations. The authors did not conduct a study of somatic health or psychological development of children in connection with the different involvement of fathers in the child's life, or in connection with attending kindergarten. The psychological development was concluded on the basis of the mother's or the caregivers's answers to questions concerning, inter alia, literacy, numeracy, socio-emotional skills and learning. The authors refer to the assessment of the collected material, among others to data from Mongolian Multiple Indicator Cluster Surveys (MICS).

Some of the "weaknesses" of this study were indicated above: limiting ourselves to data obtained from mothers and not conducting psychological (or medical) tests. At this stage of the research, it is not possible to remove these limitations. It should be noted, however, that examining young children is difficult and is generally carried out in the presence of a mother of  baby / or caregiver.

The strength of the article is statistical analysis.

The conclusions were formulated correctly due to the available data and basically correct references to the literature on changes in the labor market, access to education, changes in family life, changes in the roles of women and men - in the context of global cultural transformations, including systemic transformations in some regions of the world (including Mongolia), also the importance of pre-school education and the role of the father (and mother) in the cognitive, emotional and social development of a child.

Attendance at kindergarten was found to be correlated with the wealth and level of education of parents (model 2) and the region of residence - that is, sociodemographic factors turned out to be decisive (the age of the child also turned out to be important - which is understandable due to the processes of psychological development and physical development.)

However, a question arises: in the context of the results of statistical analyzes, it can be assumed that the Mongolian society is patriarchal?

The authors do not give a positive answer to the question whether the child's psychological development is influenced by the presence and / or involvement of the father.

However, do the data provide answers to the following questions: Is preschool education influencing the child's achievements, or is the mother's activity decisive? The presented statistical analyzes show that there is no relationship between the father's commitment and the child's developmental achievements - this shows the cultural specific of the studied population.

 It was established that the father had no influence on the child's attendance at kindergarten and on child's health condition. However, the question arises: does the presence of the father and the father's involvement influence the child's developmental achievements - through the father-mother relationship or through the mother's involvement? Due to the specificity of early childhood development, the assumption that the child's age is the factor limiting the importance of the relationship with the father cannot be rejected. This results shows specificity way of eductaion in this society.

A possible explanation of the results obtained may be the father's limited chances of being an attachment figure for his child in the event of a man's periodic separation from his family due to the specificity of work and / or geographical specificity.

It would be worth the trouble to answer some questions - for example by comparing the data on children attending kindergarten with children not attending kindergarten and with taking account the periods when their father was away from home. The research material seems to allow this.

The advantage of the article is the population to which it is dedicated. In the worldwide scientific literature, little is known about Mongolian society.

(Unfortunately, this is partly due to the attitude of some publishers or editors of scientific journals that Mongolia is far from the "center of the world" and is a small nation - so research on it will not arouse much interest among potential readers. This approach is not in line with the idea of ​​globalization. unification or egalitarianism, which may result in the marginalization of some populations. This is an approach that blocks the development of science to some extent.)

The authors' conclusions confirm - on the one hand - the global trends of changes in social roles taking place in Mongolian society, and on the other hand - the importance of economic conditions and teh strengh of cultural tradition. They indicate that this is a society in the process of being "restructured".

For the above reasons, I find the article interesting.

The authors are aware of the limitations of their research, which are sociodemographic and, to a lesser extent, psychological.

 (Note: I suggest going back to Bronfenbrenner's theory, including understanding relationships in micro-, exo-, meso -, macrosystem, and understanding chronosystem) - in texts other than the 1977 study.)

I propose to consider the title:

Father's Presence, Father's Engagement, and Some Developmental Achievements of Preschool Children in Mongolia

Or:

I propose to consider the title: Father's Presence, Father's Engagement, and Some Developmental Outcomes of Preschool Children

Author Response

Thank you for the thorough review of our paper. We have considered your comments and suggestions and address them as following:

1. It is worth paying attention to the cultural tradition of the Mongols: for example, the dominant religious systems, which may be important for gender roles, including parental ones, as well as for the expectations of social egalitarianism. Many questions arise, for example the following question: can the identified differences between regions be related to a different worldview of the inhabitants of these regions? Did the migrations in the 21st century limited these differences? or influenced on larger differences within regions?

Response: We added additional statements in the Introduction section [lines 43 – 51] and two additional paragraphs in the Discussion section [lines 493 - 526], portraying how traditions and cultures are affecting the gender role in Mongolia and any shifts in modern society and possible explanations that relates to our findings and discussion. As we reflect on the regional differences, regions and provinces have been affected by environmental changes at different rates but the regions do not have huge differences in sociocultural or philosophical differences in their views of gender and parenting roles.

2. A question arises: in the context of the results of statistical analyzes, it can be assumed that the Mongolian society is patriarchal?

Response: Mongolia remains a patriarchal society, historically stemming from traditional nomadic pastoralist households where roles were explicitly gendered. Additional information on gender roles and the corresponding changes during the transition has been detailed in the Introduction and Discussion sections [lines 43-51; lines 493-509].

3. The authors do not give a positive answer to the question whether the child's psychological development is influenced by the presence and / or involvement of the father.

Response: We now cite a meta-analysis by Jeynes (2016) that investigated the unique role of fathers in child development. This study suggests that father involvement had a positive association with psychological outcomes in their young child. This influence of father involvement has now been emphasized in the text [line 121-122].

4. Is preschool education influencing the child's achievements, or is the mother's activity decisive?

Response: We initially included ‘mother engagement’ in the multivariate model. However, it did not contribute to the model (i.e. mother engagement was not associated with preschool attendance in children), and since we were primarily interested in father engagement, ‘mother engagement’ was omitted from the final model.

5. It was established that the father had no influence on the child's attendance at kindergarten and on child's health condition. However, the question arises: does the presence of the father and the father's involvement influence the child's developmental achievements - through the father-mother relationship or through the mother's involvement?

Response: Thank you for this comment. While it is understood that family dynamics within the household are complex, we were primarily interested in investigating the father-child relationship within Mongolian households due to the concerns of shifting gender roles in a post-transitional and patriarchal society. We agree that the additional influence of the mother should also be examined and can be pursued in follow-up studies.

Preschool attendance was chosen as the outcome measure for two reasons, namely, to 1) investigate an important child outcome according to the UN’s Sustainable Development Goals, ie. “ensuring that all girls and boys have access to quality early childhood development, care, and pre-primary education so that they are ready for primary education” [as cited on lines 137-139], and 2) to investigate factors that may explain why only 68% of preschool-aged children attended preschool in 2013 [as cited on lines 148 - 149]. This was of interest particularly since Mongolian populations tend to be well-educated. The relationship between father involvement and childhood development achievements has previously been explored in an international study (e.g. Tran et al., 2016), however, we agree that this is worth investigating strictly within the Mongolian context and can be done through future studies.

6. Explanation of the results obtained may be the father's limited chances of being an attachment figure for his child in the event of a man's periodic separation from his family due to the specificity of work and / or geographical specificity. It would be worth the trouble to answer some questions - for example by comparing the data on children attending kindergarten with children not attending kindergarten and with taking account the periods when their father was away from home. The research material seems to allow this.

Response: The father’s periodic separation from the family was initially considered in light of the macro-environmental changes happening in Mongolia. Unfortunately, due to the cross-sectional nature of the MICS as well as the survey design, we could only measure whether the father was living in the same household at the time of the survey (i.e. Father Presence). The MICS 2013 questionnaires did not have any additional questions regarding the father’s length of absence or periodic absence. This limitation was added to the text on lines 552 – 555.

7. I suggest going back to Bronfenbrenner's theory, including understanding relationships in micro-, exo-, meso -, macrosystem, and understanding chronosystem) - in texts other than the 1977 study.

Response: We have now included more recent citations for Bronfenbrenner’s Ecological Theory [line 94] and have refined our definition of the ‘chronosystem” [line 102].

8. I propose a change to the title: Father's Presence, Father's Engagement, and Some Developmental Achievements of Preschool Children in Mongolia or Father's Presence, Father's Engagement, and Some Developmental Outcomes of Preschool Children

Response: Thank you for the comment we have decided to make a change to the title of our article based on your suggestions.

Reviewer 2 Report

Article Review

The article, “Father Presence, Father Engagement, and Child Outcomes in Mongolia,” presents the results of a cross-sectional analysis of the role fathers play in their children’s lives in a Mongolian cultural context. Data about three- and four-year-old children were examined from the UNICEF Multiple Indicator Cluster Survey.  The authors note that in a developing world in which the climate and economy is changing, the role that father’s play in positive outcomes of children is critical to understand.  While research on this topic has been done in many cultural contexts, none has been done in Mongolia.  Mongolia is an interesting case study because of the rapid changes that it has under the past few decades.  The authors examined paternal presence and paternal engagement and how it related to childhood acute illness and preschool attendance, while controlling for multiple confounding variables.  Overall, the researchers found no significant effects of paternal presence or engagement on child outcomes after controlling for confounding variables.  The authors conclude that these confounding variables, such as maternal education, SES, and geographical region are more important variables.

Overall Review

Overall, the article was well-written and clearly presented the results of a well-conducted study.  Only one major issue needs to be addressed, as well as a few minor issues.

Discussion

The discussion of results did a very good job explaining how the confounding variables are incredibly important for the selected child outcomes.  Additionally, the discussion of limitations was comprehensive.  However, a deeper exploration of the null findings is needed.  Null findings are inherently interesting, especially when not hypothesized.  The authors note that despite not finding a statistical link between the paternal variables and child outcomes, the reader should not take this to mean that paternal involvement is not important. This is true.  But the authors should at least explore why the null findings exist from a theoretical perspective, and not just a statistical or study design perspective.  If it is true that paternal involvement is not important in this context, it would be interesting to see suggestions for why.

Minor Issues

  • The authors note that father engagement was measured through how many of 6 activities the father completed with their children. They provide a citation to show that this is a valid way of operationalizing father engagement.  But is this a valid way in this cultural context?  Given that the authors stress the importance of understanding the Mongolian cultural context, the variable should be valid for this context.  I am not suggesting that the authors use a different variable.  Just that they should include a sentence or two explaining why it is also valid for this population.
  • For the variable of preschool attendance, answers of “Don’t Know” were recoded as “No.” An explanation of this decision is needed.
  • Given that the region of Mongolia is a statistically significant variable in the analyses, a small discussion of the different regions and why there might be differences should be included in the introduction.

Author Response

Thank you for the thorough review of our manuscript. We have responded to your specific concerns as follows:

1. The discussion of results did a very good job explaining how the confounding variables are incredibly important for the selected child outcomes.  Additionally, the discussion of limitations was comprehensive.  However, a deeper exploration of the null findings is needed.  Null findings are inherently interesting, especially when not hypothesized.  The authors note that despite not finding a statistical link between the paternal variables and child outcomes, the reader should not take this to mean that paternal involvement is not important. This is true.  But the authors should at least explore why the null findings exist from a theoretical perspective, and not just a statistical or study design perspective.  If it is true that paternal involvement is not important in this context, it would be interesting to see suggestions for why.

Response: Thank you for this comment, we agree that a more in depth exploration and discussion of our null findings was needed. We have included two additional paragraphs in the Discussion to offer some explanation for our findings [lines 493-526]. 

2. The authors note that father engagement was measured through how many of 6 activities the father completed with their children. They provide a citation to show that this is a valid way of operationalizing father engagement.  But is this a valid way in this cultural context?  Given that the authors stress the importance of understanding the Mongolian cultural context, the variable should be valid for this context.  I am not suggesting that the authors use a different variable.  Just that they should include a sentence or two explaining why it is also valid for this population.

Response: Thank you for your comment. We agree that the measure for father engagement does not fully capture parenting within the specific Mongolian context. Health indicator data from the MICS are meant to monitor a nation’s progress towards the Millennium Development Goals and now the Sustainable Development Goals with the additional goal of comparing rates across low- and middle-income countries. With this goal in mind, the MICS were designed accordingly, and the six activities in particular were meant to measure father’s support for learning. While it would be ideal to utilize questions created specifically for the Mongolian context, this was the most comprehensive dataset available at the time that collected information on father involvement. This is a study limitation and has now been better noted on line 544.

3. For the variable of preschool attendance, answers of “Don’t Know” were recoded as “No.” An explanation of this decision is needed.

Response: We wanted to retain as many cases as possible in the analysis. Therefore, in order to retain those that answered “don’t know” to preschool attendance (n=3), we decided to recode them as “no” responses. This has now been noted in the text on lines 256 and 257.

4. Given that the region of Mongolia is a statistically significant variable in the analyses, a small discussion of the different regions and why there might be differences should be included in the introduction.

Response: We appreciate the comment on exploring the regional and geographical considerations of Mongolia, we added a statement on lines 49 - 51 to explain the nature of this concern.

Reviewer 3 Report

This paper addresses a potentially important topic. There are two key themes in the paper, one about parenting and child development outcomes in Mongolia, especially in relation to a changing cultural and economic context. The other theme is about the role of fathers in child development.  The main strength of the paper is that it describes and comments on the social, economic and childrearing context in Mongolia well. The other theme of the paper, about fathers, in terms of their role in the family and childrearing is probably less well developed. The authors draw on a number of important contributors to scholarship on fathers. But the discussion was not well developed about the role of father presence and engagement as these constructs apply to the present study.  Much of this is covered over lines 121 to 132. The authors draw on the idea of “positive father involvement” and note that it pertains to accessibility, engaging with the child, and responsible for planning and managing the needs of their child. These ideas probably are most relevant in relation to authoritative parenting roles. The authors, in contrast, claim that the parenting context in Mongolia is authoritarian. Here they cite the paper of Graf et al. (2014). My reading of the Graft paper raised questions about that claim. Graf et al. seemed to provide evidence of greater use of corporal punishment, but less use of power assertion. In the end, they seemed to describe the Mongolian socialization context as one of “autonomous relatedness”. 

Given the important of cultural models for socialization goals and parenting behavior (cf the extensive work of Kagitcibasi), it is critical for the reader here to have an appreciation of the social and cultural context for these Mongolian parents and how it might influence their socialization goals and parenting behavior. It is a little complex when one considers that the present sample differs from the sample of parents that Graf et al. used. Nevertheless, it is would be helpful if the authors explained more about whether and how the parenting styles of these parents might be best described as “authoritarian” versus based on “autonomous relatedness”. Either way, this kind of explanation is important for a reader to better understand the role of fathers in relation to their participation in the six forms of engagement studied, together with the manner and purpose that fathers might have in mind in relation to these forms of engagement. Do fathers engage in these activities as a part of goals of relatedness, or might they engage in them as part of goals of increasing the child’s school readiness? If it is the former, this could be a factor in why there were no relationships between father engagement and preschool attendance. One could also ask if the father engagement is about school readiness why the outcome measure is attendance and not something about outcomes linked more closely to readiness.

I had some questions about the results. The research found that father presence and engagement were not associated with child illness or pre-school attendance. It looked like 1896 of the sample of 2220 were present. In terms of engagement, the results in Figure 2 indicate that less than 20% of fathers were engaged in most of these activities at some time over the 3 days that mothers were asked about. These results raise the question of ceiling (the presence measure) and floor (the engagement measure) effects in the data. It is not clear whether or to what extent these activities could be indicative of positive father involvement. For example, they could be conducted in a formal manner rather than in terms of the positive involvement that the authors have taken from the literature in other cultural contexts. It would really help a reader to understand the “engagement” measure if it was discussed in relation to the broader role of parents in Mongolia, especially in relation to the claimed authoritarian role of fathers (or autonomous relatedness) and gender roles and patriarchal cultural values that the authors mentioned.

The authors draw on Bronfenbrenner’s ecological theory of development that is widely cited in the Western literature. Again, there seems a need to explain how this model could be adapted to the Mongolian context. In particular, how the various systems that form part of the model would apply in Mongolia and as a consequence, the effects of these systems on interactions in the microsystem of parent-child interactions. 

The authors make an important claim in lines 133 and 134, that knowledge about fathers will help understand factors associated with child health and well-being.  In the next two paragraphs the authors provide helpful literature about results from low and middle-income countries.  The level of income is relevant to the argument, but is the cultural context (such as authoritarian parenting, patriarchal values or autonomous relatedness) also relevant? Subsequently, when discussing the results of the present study, it would be helpful for the authors to consider the measures they used against the measures used in earlier research where father presence and engagement/involvement were examined. How well did the present measures tap into something like positive father involvement for example?

I didn’t find the last paragraph of the introduction as helpful as it might be. You make a general claim about investigating the father “role”.  Rather, the research is about simple presence and about some very specific ratings of father activities with their child.  Being more specific about the purpose of the research here would help.

I wasn’t sure on line 246 what was meant by “any of the activities”.  I got the impression that mothers were asked about eachof the activities.

I had some questions about the information in Table 1. For example, why the early childhood development score and books in the family were listed only under father engagement.

Overall, this paper is well-written. The methods, data analysis and results are clearly explained. Many aspects of the Mongolian context are very helpfully covered. This contributes to the significance of the paper in relation to the international literature on fathers.  It would help, however, if the social, cultural (and possibly religious) context were better described, so that a reader could be helped to understand more about the roles of mothers and fathers and therefore better understand the nature and possible contribution of the specific measures of engagement used in the research.

A complication is that the paper is already rather long when considered against (1) the possible limitations of the measure of father engagement as an indicator of father involvement, (2) the findings (where the possible limitations of the measures of father engagement could be a factor), and (3) possible design limitations (for example, the selection of father engagement as a possible influence on preschool attendance rather than as an influence on a more proximal outcome).

Author Response

Thank you for the thorough review of our manuscript. We have addressed your concerns in the following ways:

1. I didn’t find the last paragraph of the introduction as helpful as it might be. You make a general claim about investigating the father “role”.  Rather, the research is about simple presence and about some very specific ratings of father activities with their child.  Being more specific about the purpose of the research here would help.

Response: Thank you for bringing this to our attention. This has been updated on lines 181-185 to precisely reflect the purpose of this study.

2. I wasn’t sure on line 246 what was meant by “any of the activities”.  I got the impression that mothers were asked about each of the activities.

Response: Thank you for your comment. This has been clarified on line 232-233.

3. I had some questions about the information in Table 1. For example, why the early childhood development score and books in the family were listed only under father engagement.

Response: Thank you for bringing this to our attention. The mean for ‘number of books’ has been added to Table 1. Early Childhood Development Score was not included since the effect of father presence on ECD score would most likely occur through father engagement with the child. Therefore, this variable was only included in the Father Engagement model.

4. It would help, if the social, cultural (and possibly religious) context were better described, so that a reader could be helped to understand more about the roles of mothers and fathers and therefore better understand the nature and possible contribution of the specific measures of engagement used in the research.

Response: We appreciate the reviewer for pointing out the contexts of gendered roles in parenting. We added a few additional sentences in the Introduction emphasizing macro environmental changes occurring in Mongolia and specific problems that are relevant to findings of our study [lines 43 – 51]. We also added two additional paragraphs starting on line 493 in the Discussion section that explains the context of social, cultural, and religious aspects of parenting in Mongolia.

5. It would be helpful for the authors to consider the measures they used against the measures used in earlier research where father presence and engagement/involvement were examined. How well did the present measures tap into something like positive father involvement for example?

Response: Similar dichotomous measures of father presence have been used in prior studies such as that by Schmeer 2009. In their study, father presence was measured as whether the father currently lived in the household or if the father was absent due to migration. Other studies (e.g. Ziol-Guest & Dunifon, 2014) measured father presence using information on family structures, such as whether the child lived with a single parent or step-parent. In our study, while we were able to capture father presence in the household, we were limited to the amount of information available regarding the father’s periodic absence or length of absence, if any. This limitation was noted in lines 552-553.

As for father engagement/involvement, prior studies in low- and middle-income countries used the same measures in the MICS to examine the relationship between father involvement/father support for learning and child development. Studies such as those cited in the manuscript (e.g. Jeong et al., 2016; Jeong et al., 2017) have consistently shown that higher levels of father engagement were associated with more positive outcomes in the child. However, we are aware that the MICS does not capture other forms of involvement such as feeding or bathing the child, nor does it capture the quality of these interactions, mainly because measures of parental involvement were designed to capture support for learning. However, it is worth nothing that this was the most comprehensive dataset available at the time that had some measure of father involvement, and therefore, follow-up studies are warranted once additional measures are available.

6. The authors draw on Bronfenbrenner’s ecological theory of development that is widely cited in the Western literature. Again, there seems a need to explain how this model could be adapted to the Mongolian context. In particular, how the various systems that form part of the model would apply in Mongolia and as a consequence, the effects of these systems on interactions in the microsystem of parent-child interactions. 

Response: We applied the Bronfenbrenner’s ecological theory in Mongolia context, and added clarifications on how this theory relates to a non-western nomadic culture in lines 103 – 105 and lines 510-526, visualizing how father’s involvements are noticed.

7. It would really help a reader to understand the “engagement” measure if it was discussed in relation to the broader role of parents in Mongolia, especially in relation to the claimed authoritarian role of fathers (or autonomous relatedness) and gender roles and patriarchal cultural values that the authors mentioned.

Response: We have added two additional paragraphs to the Discussion section to provide a more detailed description of parenting within the Mongolian context [lines 493 – 526].

8. Given the important of cultural models for socialization goals and parenting behavior (cf the extensive work of Kagitcibasi), it is critical for the reader here to have an appreciation of the social and cultural context for these Mongolian parents and how it might influence their socialization goals and parenting behavior...Here they cite the paper of Graf et al. (2014). ...Graf et al. seemed to provide evidence of greater use of corporal punishment, but less use of power assertion. In the end, they seemed to describe the Mongolian socialization context as one of “autonomous relatedness”. 

Response: Thank you for bringing these points to our attention. We agree that parenting practices are cultural and social constructs and have, given your suggestion, cited Kagitcibasi’s work to support this statement. In addition, we included two additional paragraphs in the Discussion [lines 493 – 526] to provide the readers with relevant historical, cultural, and social contexts that shape Mongolian parenting behaviours.

In terms of the type of parenting, we have now updated this statement on lines 72-74 to better explain Mongolian father involvement within the context of autonomous relatedness.

9. One could also ask if the father engagement is about school readiness why the outcome measure is attendance and not something about outcomes linked more closely to readiness.

Response: Preschool attendance was chosen as the outcome measure for two reasons, namely, to 1) investigate an important child outcome according to the UN’s Sustainable Development Goals, ie. “ensuring that all girls and boys have access to quality early childhood development, care, and pre-primary education so that they are ready for primary education” [as cited on lines 137-139], and 2) to investigate factors that may explain why only 68% of preschool-aged children attended preschool in 2013 [as cited on lines 148-149]. This was of interest particularly since Mongolian populations tend to be well-educated.

Round 2

Reviewer 3 Report

The authors have done a good job in responding to my comments and suggestions, apart from one matter. This is my question about the distribution of the data (possible ceiling and floor effect) and whether this might have contributed to the lack of positive results. Is there a need to say something about this in the paper? 

A minor matter: the information on lines 43-51 is duplicated to a considerable degree in the discussion from line 498.

Author Response

Thank you for these two additional comments. We have edited the manuscript as follows:

  1. We have edited the text around lines 498 (now 503-510) to ensure that they are not duplicative from lines 43-51;
  2. We agree that there is a possibility of a ceiling/scale attenuation effect and we have now added mention of this in lines 553-558.